# Therapeutic Potential of an Anti-CD44v6 Monoclonal Antibody in Xenograft Models of Colorectal and Gastric Cancer

**DOI:** 10.3390/cells14231873

**Published:** 2025-11-26

**Authors:** Aoi Hirayama, Tomohiro Tanaka, Tomokazu Ohishi, Keisuke Shinoda, Takuya Nakamura, Airi Nomura, Naoki Kojo, Haruto Araki, Kaito Suzuki, Mika K. Kaneko, Hiroyuki Suzuki, Yukinari Kato

**Affiliations:** 1Department of Antibody Drug Development, Tohoku University Graduate School of Medicine, 2-1 Seiryo-machi, Aoba-ku, Sendai 980-8575, Japan; hirayama.aoi.p7@dc.tohoku.ac.jp (A.H.); tomohiro.tanaka.b5@tohoku.ac.jp (T.T.); shinoda.keisuke.q3@dc.tohoku.ac.jp (K.S.); nakamura.takuya.r4@dc.tohoku.ac.jp (T.N.); nomura.airi.p3@dc.tohoku.ac.jp (A.N.); kojo.naoki.q4@dc.tohoku.ac.jp (N.K.); araki.haruto.r8@dc.tohoku.ac.jp (H.A.); suzuki.kaito.q1@dc.tohoku.ac.jp (K.S.); mika.kaneko.d4@tohoku.ac.jp (M.K.K.); 2Laboratory of Oncology, Institute of Microbial Chemistry (BIKAKEN), Microbial Chemistry Research Foundation, 3-14-23 Kamiosaki, Shinagawa-ku, Tokyo 141-0021, Japan; ohishit@bikaken.or.jp

**Keywords:** monoclonal antibody therapy, CD44v6, ADCC, CDC, gastric cancer, colorectal cancer

## Abstract

CD44 variant (CD44v) isoforms are involved in promoting cancer metastasis, sustaining cancer stem cell (CSC) properties, and conferring resistance to therapeutic interventions. Consequently, the development of monoclonal antibodies (mAbs) targeting CD44v represents a crucial strategy for eliminating CD44v-positive cancer cells. Previously, an anti-CD44v6 mAb, C_44_Mab-9 (mouse IgG_1_, κ), was established. C_44_Mab-9 recognizes explicitly the epitope encoded by the variant exon 6-encoded region of CD44 and applies to flow cytometry, western blotting, and immunohistochemistry. To assess the therapeutic potential, a mouse IgG_2a_ isotype of C_44_Mab-9 (designated C_44_Mab-9-mG_2a_) was generated, and the in vitro and in vivo antitumor activities were evaluated using gastric and colorectal cancer cell lines. C_44_Mab-9-mG_2a_ demonstrated specific binding to CD44v3–10-overexpressed Chinese hamster ovary cells (CHO/CD44v3–10), as well as gastric cancer (NUGC-4) and colorectal cancer (COLO201 and COLO205) in flow cytometry. C_44_Mab-9-mG_2a_ exerted antibody-dependent cellular cytotoxicity (ADCC) and complement-dependent cytotoxicity (CDC) against CHO/CD44v3–10, NUGC-4, COLO201, and COLO205. Moreover, systemic administration of C_44_Mab-9-mG_2a_ significantly inhibited tumor growth in CHO/CD44v3–10, NUGC-4, COLO201, and COLO205 xenografts compared with the control IgG_2a_. These findings indicate that C_44_Mab-9-mG_2a_ could be applied to the mAb-based therapy against CD44v6-positive tumors.

## 1. Introduction

CD44 was first characterized as a lymphocyte homing receptor and plays crucial roles in cell migration, adhesion, survival, proliferation, and stemness [1]. The human CD44 gene comprises multiple exons, with exons 1–5 encoding the conserved extracellular domain, exons 16 and 17 encoding the stem region, exon 18 encoding the transmembrane domain, and exons 19 and 20 encoding the cytoplasmic domain. The isoform containing exons 1–5 and 16–20 is designated as the standard form (CD44s), which is ubiquitously expressed across most cell types. In contrast, alternative splicing of exons 6–15 gives rise to multiple CD44 variant (CD44v) isoforms, which are inserted between the extracellular and stem regions [2]. For example, CD44v8–10 (also termed CD44E) is primarily expressed in epithelial cells, whereas CD44v3–10 is expressed in keratinocytes [3]. Dysregulated expression of CD44v isoforms has been implicated in tumor progression and metastasis [4].

Pan-CD44, including CD44s and CD44v, interacts with hyaluronic acid (HA) via the conserved extracellular domain, which plays pivotal roles in cellular proliferation, migration, homing, and adhesion [5]. The CD44v isoforms possess diverse oncogenic functions, including the promotion of cancer cell motility and invasiveness [6], acquisition of cancer stem cell (CSC)-like properties [4], and resistance to radiotherapy and chemotherapy [7]. Each variant exon-encoded region plays an essential role in cancer progression. The v3-encoded region undergoes modification of heparan sulfate, enabling the recruitment of heparin-binding growth factors, including fibroblast growth factors. Therefore, the v3 region can amplify the downstream signaling of receptor tyrosine kinases [8]. The v6-encoded region is critical for activation of c-MET via a ternary complex formation with the ligand, hepatocyte growth factor (HGF) [9]. Moreover, the v8–10-encoded region interacts with xCT, a cystine–glutamate transporter, and plays pivotal roles in the reduction of oxygen species through promotion of cystine uptake and glutathione synthesis [10].

CSCs drive tumor initiation and metastasis [11,12]. CD44 has been implicated as a cell surface CSC marker in various carcinomas [3]. In colorectal cancer, the CSCs express CD44v6, which is essential for their migration and metastasis [13]. Although CD44v6 expression is relatively low in the primary tumors, it distinctly marks the clonogenic CSC subpopulation [14]. Cytokines such as HGF derived from the tumor microenvironment, upregulate CD44v6 expression in the CSCs via activation of the Wnt/β-catenin signaling pathway, thereby promoting migration and metastasis [14]. In patient cohorts, CD44v6 expression is an independent negative prognostic marker [14]. Therefore, CD44v6 has been implicated as an essential target for tumor diagnosis and therapy.

Anti-CD44v6 monoclonal antibodies (mAbs), including clones VFF4, VFF7, and VFF18, have been developed and primarily utilized for tumor diagnosis and therapeutic applications. The VFF series of mAbs were generated by immunizing animals with bacterially expressed CD44v3–10 fused to glutathione S-transferase [15,16]. Among them, VFF4 and VFF7 were employed in immunohistochemical analysis [17]. VFF18 was shown to bind specifically to fusion proteins, containing the variant 6-encoded region. Moreover, enzyme-linked immunosorbent assay (ELISA) using synthetic peptides spanning this region identified the WFGNRWHEGYR sequence as the VFF18 epitope [15].

VFF18 (mouse IgG_1_) was further humanized and developed to BIWA-4 (bivatuzumab) [18]. Clinical trials of the BIWA-4 conjugated with mertansine (bivatuzumab mertansine) were conducted for the treatment of solid tumors. But they were discontinued due to skin toxicities [19,20]. These adverse effects were attributed to the accumulation of the mertansine payload in normal skin squamous epithelium [19,20]. In addition, human acute myeloid leukemia (AML) cells have been shown to express high levels of CD44 mRNA as a result of reduced CpG island methylation within the promoter region [21]. Notably, elevated CD44v6 expression was detected in AML patients harboring FLT3 or DNMT3A mutations [21]. Consequently, a modified version of BIWA-4, designated BIWA-8, was developed to generate chimeric antigen receptors (CARs) targeting CD44v6 in AML. These CD44v6-directed CAR-T cells demonstrated strong anti-leukemic activity [21,22], suggesting that CD44v6 is a promising therapeutic target for AML with FLT3 or DNMT3A mutations. Moreover, CD44v6 CAR-T cells also exhibited significant antitumor efficacy in xenograft models of lung and ovarian cancers [23]. The safety and effectiveness of CD44v6 CAR-T treatment in patients with advanced CD44v6-positive solid tumors are being evaluated in Phase I/II clinical studies [24]. Moreover, BIWA-8-based CAR-NK therapy also showed effectiveness in eliminating CD44v6-positive cancer cell lines [25]. Accordingly, VFF18-derived BIWA-4 or BIWA-8-based therapies have been evaluated in the clinic.

We previously generated a series of highly sensitive and specific mAbs against CD44v by immunizing mice with Chinese hamster ovary-K1 cells stably overexpressed CD44v3–10 (CHO/CD44v3–10). The critical epitopes recognized by these mAbs were determined by ELISA and further characterized through flow cytometry, Western blotting, and immunohistochemistry [26]. Among the established clones, C_44_Mab-9 (mouse IgG_1_, κ) specifically bound to a peptide corresponding to the v6-encoded region [26]. Flow cytometric analysis revealed that C_44_Mab-9 recognized both CHO/CD44v3–10 cells and colorectal cancer cell lines. Moreover, C_44_Mab-9 successfully detected endogenous CD44v6 in colorectal cancer tissues by immunohistochemistry. Collectively, these findings indicate that C_44_Mab-9 is a valuable mAb for detecting CD44v6 in a variety of experimental and diagnostic applications.

In the present study, we engineered a mouse IgG_2a_ version of C_44_Mab-9 (designated C_44_Mab-9-mG_2a_) to enhance effector functions. We then evaluated its antibody-dependent cellular cytotoxicity (ADCC), complement-dependent cytotoxicity (CDC), and antitumor efficacy in xenograft models of gastric and colorectal cancers.

## 2. Materials and Methods

### 2.1. Cell Lines

CHO-K1 and colorectal cancer cell lines (COLO201 and COLO205) were obtained from the American Type Culture Collection (Manassas, VA, USA). A human gastric cancer cell line (NUGC-4) was obtained from the Japanese Collection of Research Bioresources (Osaka, Japan). These cells and CHO/CD44v3–10 were maintained as described previously [26].

### 2.2. Recombinant mAb Production

An anti-CD44v6 mAb, C_44_Mab-9 (mouse IgG_1_, κ) was previously established [26]. To create the mouse IgG_1_ version (C_44_Mab-9) and mouse IgG_2a_ version (C_44_Mab-9-mG_2a_), the V_H_ cDNA and V_L_ cDNA of C_44_Mab-9 were cloned and generated the expression vectors as reported previously [27]. The antibody production and purification were performed using the ExpiCHO Expression System (Thermo Fisher Scientific Inc., Waltham, MA, USA), as described previously [27]. C_44_Mab-9 and C_44_Mab-9-mG_2a_ were purified using Ab-Capcher (ProteNova Co., Ltd., Kyoto, Japan). A control mouse IgG_2a_ (mIgG_2a_) mAb, PMab-231 (mouse IgG_2a_, κ, an anti-tiger podoplanin mAb) was previously produced [26]. PMab-231 and C_44_Mab-9-mG_2a_ were denatured by SDS sample buffer (Nacalai Tesque, Inc., Kyoto, Japan) containing 2-mercaptoethanol and subject to SDS-PAGE. The gel was stained with Bio-Safe CBB G-250 Stain (Bio-Rad Laboratories, Inc., Berkeley, CA, USA).

### 2.3. ELISA

Twenty synthesized CD44v6 peptides [CD44v6 p351-370 wild-type (WT) and the alanine-substituted mutants] were synthesized by Sigma-Aldrich Corp., St. Louis, MO, USA. ELISA was performed as described previously [26].

### 2.4. Flow Cytometry

Cells, collected by brief incubation with a solution containing 1 mM ethylenediaminetetraacetic acid (EDTA; Nacalai Tesque, Inc.) and 0.25% trypsin, were washed with a blocking buffer (PBS containing 0.1% BSA). The cells were treated with primary mAbs for 30 min at 4 °C and were incubated with Alexa Fluor 488-conjugated anti-mouse IgG (1:2000; Cell Signaling Technology, Inc., Danvers, MA, USA, AB_10694704). The data were obtained using an SA3800 Cell Analyzer (Sony Corp., Tokyo, Japan). The dissociation constant (*K*_D_) value was also determined by flow cytometry [26].

### 2.5. Measurement of ADCC by C_44_Mab-9-mG_2a_

The Institutional Committee for Animal Experiments of the Institute of Microbial Chemistry approved the animal experiment (approval numbers: 2025-029 and 2025-059). Female BALB/c nude mice (five weeks old) were purchased from Japan SLC Inc. (Shizuoka, Japan). The antibody-dependent cellular cytotoxicity (ADCC) activity of C_44_Mab-9-mG_2a_ was assessed as follows: Calcein AM-labeled target cells (CHO/CD44v3–10, NUGC-4, COLO201, and COLO205) were co-incubated with effector splenocytes at an effector-to-target (E:T) ratio of 50:1 in the presence of either 100 μg/mL C_44_Mab-9-mG_2a_ (n = 3) or control mIgG_2a_ (n = 3). After a 4.5-h incubation, Calcein release into the medium was measured, and cytotoxicity was calculated as previously described [27].

### 2.6. Measurement of CDC by C_44_Mab-9-mG_2a_

Calcein AM-labeled target cells (CHO/CD44v3–10, NUGC-4, COLO201, and COLO205) were seeded and incubated with rabbit complement (final concentration, 10%; Low-Tox-M Rabbit Complement, Cedarlane Laboratories, Hornby, ON, Canada) in the presence of 100 μg/mL C_44_Mab-9-mG_2a_ (n = 3) or control mIgG_2a_ (n = 3). After incubation for 4.5 h at 37 °C, Calcein release into the medium was quantified as previously described [27].

### 2.7. Antitumor Activity of C_44_Mab-9-mG_2a_

The Institutional Committee for Animal Experiments of the Institute of Microbial Chemistry approved the animal experiment (approval numbers: 2025-011 and 2025-029). Throughout the study, mice were maintained under specific pathogen-free conditions with an 11-h light/13-h dark cycle and were given food and water ad libitum. Body weight was monitored twice weekly, and overall health status was evaluated three times per week. Humane endpoints were defined as either a loss of more than 25% of the initial body weight and/or a tumor volume exceeding 3000 mm^3^.

Tumor cells were mixed with BD Matrigel Matrix Growth Factor Reduced (BD Biosciences, San Jose, CA, USA). Subsequently, 100 μL of the mixture, corresponding to 5 × 10^6^ cells, was subcutaneously injected into the left flank of female BALB/c nude mice (4 weeks old; Japan SLC, Inc., Shizuoka, Japan) (day 0). To assess the antitumor activity of C_44_Mab-9-mG_2a_ against CHO/CD44v3–10 and NUGC-4 tumor-bearing mice, they received intraperitoneal injections of 200 μg C_44_Mab-9-mG_2a_ (n = 8) or control mIgG_2a_ (n = 8) diluted in 100 μL PBS on days 7 and 14. Mice were euthanized on day 21 after tumor inoculation. In COLO201 and COLO205 tumor-bearing mice, 100 μg C_44_Mab-9-mG_2a_ (n = 8) or control mIgG_2a_ (n = 8) were intraperitoneally injected on days 7 and 14. Mice were euthanized on day 21 after tumor inoculation.

Tumor size was measured, and volume was calculated as previously described [27]. Data are shown as the mean ± SEM. Statistical analysis was performed using one-way ANOVA followed by Sidak’s post hoc test. A *p*-value < 0.05 was considered statistically significant.

## 3. Results

### 3.1. Production of Recombinant Anti-CD44v6 mAbs

C_44_Mab-9, an anti-CD44v6 mAb, was previously established by immunizing mice with CHO/CD44v3–10. C_44_Mab-9 recognizes the CD44 variant exon 6-encoded region determined by ELISA. C_44_Mab-9 shows a high binding affinity against CHO/CD44v3–10 and tumor cells in flow cytometry [26]. In this study, the complementarity-determining regions (CDRs) of C_44_Mab-9 were first determined from the cDNA of C_44_Mab-9-producing hybridoma. To produce a recombinant C_44_Mab-9, the V_H_ and V_L_ CDRs of C_44_Mab-9 were identified and fused with the C_H_ and C_L_ chains of mouse IgG_1_ (Figure 1A). We confirmed that a recombinant C_44_Mab-9 recognized the CD44v6 p351–370 (WT) peptide within the CD44 variant exon 6-encoded region. Furthermore, the reactivity of the alanine-substituted mutants by C_44_Mab-9 revealed that E358, W360, F361, G362, R364, and W365 are essential amino acids as the C_44_Mab-9 epitope (Figure 1B). Next, a mouse IgG_2a_-type C_44_Mab-9 (C_44_Mab-9-mG_2a_) was constructed by fusing the V_H_ and V_L_ CDRs of C_44_Mab-9 with the C_H_ and C_L_ chains of mouse IgG_2a_ to evaluate the antitumor activity (Figure 1A). As a control mouse IgG_2a_ (mIgG_2a_), PMab-231 (an anti-tiger podoplanin mAb, mouse IgG_2a_) was also produced [28]. We confirmed the purity of the recombinant mAbs by SDS-PAGE under reduced conditions (Figure 1C).

### 3.2. Flow Cytometry Using C_44_Mab-9-mG_2a_

The reactivity of C_44_Mab-9-mG_2a_ was confirmed using CHO/CD44s and CHO/CD44v3–10. C_44_Mab-9-mG_2a_ exhibited a dose-dependent reactivity to CHO/CD44v3–10, but not to CHO/CD44s or CHO-K1 (Figure 2A). Furthermore, the expression of CD44s was confirmed by an anti-pan-CD44 mAb, C_44_Mab-46-mG_2a_ (Figure 2B). Next, the binding affinity was investigated. The dissociation constant (*K*_D_) value of C_44_Mab-9-mG_2a_ for CHO/CD44v3–10 was determined to be 6.0 × 10^−10^ M (Figure 2C). These results indicated that C_44_Mab-9-mG_2a_ possesses comparable reactivity and affinity with parental mAb, C_44_Mab-9, as reported previously (*K*_D_: 8.1 × 10^−9^ M [26]).

C_44_Mab-9 could react with colorectal cancer cell lines, including COLO201 and COLO205, in flow cytometry [26]. The reactivity of C_44_Mab-9-mG_2a_ to gastric cancer NUGC-4 was next investigated in flow cytometry. As shown in Figure 2D, C_44_Mab-9-mG_2a_ exhibited reactivity to COLO201, COLO205, and NUGC-4. These results indicated that C_44_Mab-9-mG_2a_ possesses reactivity to CD44v6-positive cells.

### 3.3. ADCC, CDC, and Antitumor Effects Against CHO/CD44v3–10 by C_44_Mab-9-mG_2a_

The ADCC caused by C_44_Mab-9-mG_2a_ against CHO/CD44v3–10 was investigated. The splenocytes derived from BALB/c nude mice were used as an effector. C_44_Mab-9-mG_2a_ showed the ADCC against CHO/CD44v3–10 (21.7% vs. 8.2% cytotoxicity of control mIgG_2a_, *p* < 0.05, Figure 3A). No significant ADCC was observed in antigen-negative CHO-K1 (Appendix A). Next, the CDC caused by C_44_Mab-9-mG_2a_ and complement against CHO/CD44v3–10 was examined. C_44_Mab-9-mG_2a_ showed significant CDC against CHO/CD44v3–10 (37.1% vs. 19.9% cytotoxicity of control mIgG_2a_, *p* < 0.05, Figure 3B). No significant CDC was observed in antigen-negative CHO-K1 (Appendix A). The antitumor effect of C_44_Mab-9-mG_2a_ on CHO/CD44v3–10 xenografts was investigated. Following the inoculation of CHO/CD44v3–10, C_44_Mab-9-mG_2a_ or control mIgG_2a_ was intraperitoneally administered into CHO/CD44v3–10 xenograft-bearing mice on days 7 and 14. The tumor volume was measured on days 7, 10, 14, 17, and 21 after the inoculation. The C_44_Mab-9-mG_2a_ administration resulted in a potent and significant reduction in CHO/CD44v3–10 xenografts on days 14 (*p* < 0.05), 17 (*p* < 0.01), and 21 (*p* < 0.01) compared with that of control mIgG_2a_ (Figure 3C). Significant decreases in xenograft weight caused by C_44_Mab-9-mG_2a_ were observed in CHO/CD44v3–10 xenografts (70% reduction; *p* < 0.05; Figure 3D). Body weight loss was not observed in the xenograft-bearing mice treated with C_44_Mab-9-mG_2a_ (Figure 3E).

### 3.4. ADCC and CDC Against COLO201, COLO205, and NUGC-4 by C_44_Mab-9-mG_2a_

The ADCC caused by C_44_Mab-9-mG_2a_ against endogenous CD44v6-positive COLO201, COLO205, and NUGC-4 was investigated. The splenocytes derived from BALB/c nude mice were also used as an effector. C_44_Mab-9-mG_2a_ showed potent ADCC against COLO201 (32.0% vs. 10.9% cytotoxicity of control mIgG_2a_, *p* < 0.05, Figure 4A), COLO205 (19.9% vs. 4.3% cytotoxicity of control mIgG_2a_, *p* < 0.05, Figure 4B), and NUGC-4 (28.6% vs. 9.4% cytotoxicity of control mIgG_2a_, *p* < 0.05, Figure 4C). Next, the CDC caused by C_44_Mab-9-mG_2a_ against COLO201, COLO205, and NUGC-4 was examined. C_44_Mab-9-mG_2a_ showed significant CDC against COLO201 (54.2% vs. 21.5% cytotoxicity of control mIgG_2a_, *p* < 0.05, Figure 4D), COLO205 (23.7% vs. 9.1% cytotoxicity of control mIgG_2a_, *p* < 0.05, Figure 4E), and NUGC-4 (24.1% vs. 12.6% cytotoxicity of control mIgG_2a_, *p* < 0.05, Figure 4F). These results indicated that C_44_Mab-9-mG_2a_ exerts antitumor efficacy through activation of effector cells and complements in vitro.

### 3.5. Antitumor Effects by C_44_Mab-9-mG_2a_ Against COLO201, COLO205, and NUGC-4 Xenografts

The antitumor effects of C_44_Mab-9-mG_2a_ on COLO201, COLO205, and NUGC-4 xenografts were evaluated. Following the inoculation of the cell lines, C_44_Mab-9-mG_2a_ or control mIgG_2a_ was intraperitoneally administered into the xenograft-bearing mice on days 7 and 14. The tumor volume was measured on days 7, 10, 14, 17, and 21 after the inoculation. The C_44_Mab-9-mG_2a_ administration resulted in a significant reduction in COLO201 xenografts on days 14 (*p* < 0.01), 17 (*p* < 0.01), and 21 (*p* < 0.01) compared with that of control mIgG_2a_ (Figure 5A). The significant reductions were observed in COLO205 xenografts on days 10 (*p* < 0.05), 14 (*p* < 0.05), 17 (*p* < 0.01), and 21 (*p* < 0.01) compared with those of control mIgG_2a_ (Figure 5B). The significant reductions were also observed in NUGC-4 xenografts on day 21 (*p* < 0.01) compared with that of control mIgG_2a_ (Figure 5C).

Significant decreases in xenograft weight caused by C_44_Mab-9-mG_2a_ were observed in COLO201 xenografts (20% reduction; *p* < 0.05; Figure 5D), COLO205 xenografts (23% reduction; *p* < 0.05; Figure 5E), and NUGC-4 xenografts (33% reduction; *p* < 0.01; Figure 5F). Body weight loss was not observed in the xenograft-bearing mice treated with C_44_Mab-9-mG_2a_ (Figure 5G–I).

## 4. Discussion

Among the CD44v isoforms, CD44v6 plays a pivotal role in cancer progression through binding to HGF, osteopontin, and other key cytokines secreted in the tumor microenvironment [29]. In the presence of HGF, CD44v6 forms a ternary complex with c-MET, thereby enhancing the intracellular signaling [9]. Moreover, increased MUC5AC expression was observed in breast cancer brain metastasis through c-MET signaling. MUC5AC promotes brain metastasis by interacting with the c-MET and CD44v6. Blocking the MUC5AC/c-MET/CD44v6 complex with a blood–brain barrier-permeable c-MET inhibitor, bozitinib, effectively reduces MUC5AC expression and decreases the metastatic potential of breast cancer cells to the brain [30]. Additionally, CD44v6 expression was detected in a highly tumorigenic colorectal CSC population with metastatic potential [14] and gastric CSCs [31]. Therefore, CD44v6 has been considered as an attractive target for mAb-based therapy. In this study, C_44_Mab-9-mG_2a_, a mouse IgG_2a_ version of a novel anti-CD44v6 mAb, was developed and evaluated. C_44_Mab-9-mG_2a_ exhibited a superior reactivity to CD44v6-positive colorectal and gastric cancer cell lines in flow cytometry (Figure 2) and showed the antitumor activities to CHO/CD44v3–10 (Figure 3) and colorectal and gastric cancers (Figure 4 and Figure 5). These findings highlight the therapeutic efficacy of C_44_Mab-9-mG_2a_ and suggest the possibility of clinical application. We have already prepared the humanized version of C_44_Mab-9 and will evaluate the effectiveness in vivo. Additionally, we plan to develop antibody-drug conjugates (ADCs), such as deruxtecan.

In clinical trials using anti-CD44v6 mAbs, a clone VFF18-derived humanized mAb BIWA-4 was developed into an ADC, bivatuzumab mertansine [19,20]. BIWA-8, a modified version of BIWA-4, was developed to CD44v6-directed CAR-T [21,22], which has been evaluated in AML (ClinicalTrials.gov NCT04097301 [32]) and solid tumors (ClinicalTrials.gov NCT04427449). As shown in Figure 1B, the critical amino acids of C_44_Mab-9 epitope include E358, W360, F361, G362, R364 and W365, which overlap the epitope sequence (360-WFGNRWHEGYR-370) of VFF18 [15]. Since an alanine-substituted peptide at E358 reduced the reactivity of C_44_Mab-9 (Figure 1B), C_44_Mab-9 possesses a different epitope from VFF18. Furthermore, VFF18 was generated by immunization with bacterially expressed CD44v3–10 [16], whereas C_44_Mab-9 was generated by immunization with CHO/CD44v3–10 [26]. It is essential to compare the biological activity of mAbs derived from C_44_Mab-9 and VFF18.

CD44v isoforms were markedly upregulated and associated with poor prognosis in various cancers. Expression of CD44v isoforms is reported to be regulated by the Wnt-β-catenin/TCF4 signaling pathway in colorectal cancer [33]. A recent study demonstrated a novel tumor-suppressive mechanism of MEN1 through the regulation of the CD44 alternative splicing. Loss of Men1 significantly accelerated the progression of Kras-driven lung adenocarcinoma and enhanced the accumulation of CD44v isoforms [34]. Mechanistically, MEN1 maintained a relatively slow RNA polymerase II elongation rate by regulating the release of PAF1 from CD44 pre-mRNA, thereby preventing the inclusion of variant exons of CD44 [34]. Moreover, CD44v6-interfering peptides effectively inhibited the growth and metastasis of established MEN1-deficient tumors through the activation of ferroptosis, an iron-dependent form of cell death characterized by the accumulation of lipid peroxides on cellular membranes [34]. The CD44v6-interfering peptide was previously reported as KEQWFGNRWHEGYR [35], which contains the epitope amino acids of C_44_Mab-9 (Figure 1B). Furthermore, the CD44v6-interfering peptide or VFF18 were also reported to block ligand-dependent activation of c-MET and subsequent colorectal cancer cell scattering and migration [35,36]. Although we mainly investigated the ADCC and CDC induced by C_44_Mab-9-mG_2a_ (Figure 3 and Figure 4), it is interesting to examine whether C_44_Mab-9 suppresses the ligand-induced cell proliferation and induces ferroptosis in vitro and in vivo.

Because CD44 plays a pivotal role in promoting tumor metastasis and conferring resistance to therapy, multiple therapeutic strategies have been designed to target CD44 against various cancers, including breast, head and neck, ovarian, and gynecological cancers [37]. Despite these efforts, clinical trials assessing CD44-directed approaches have yielded modest results. For instance, RG7356, a mAb against pan-CD44, showed an acceptable safety profile but failed to elicit a dose-dependent or clinically meaningful response, leading to early termination [38,39,40,41]. Bivatuzumab mertansine underwent clinical testing; however, its development was discontinued due to the occurrence of severe skin toxicity [19,20].

To minimize off-target reactivity and reduce toxicity in normal tissues, cancer-specific monoclonal antibodies (CasMabs) targeting various tumor-associated antigens have been generated. Over three hundred anti-human epidermal growth factor receptor 2 (HER2) mAb clones have been established through immunization of mice with cancer cell-expressed HER2. Among these, H_2_CasMab-2 (also known as H_2_Mab-250) was identified as exhibiting cancer cell-specific binding through flow cytometric screening. H_2_CasMab-2 selectively recognized HER2 expressed on breast carcinoma cells, while showing negligible reactivity with HER2 on normal epithelial cells, derived from mammary gland, colon, lung bronchus, renal proximal tubule, and colon [42]. The epitope analyses further elucidated the molecular basis underlying this cancer-specific recognition [42,43]. In addition, CAR T-cell therapy employing a single-chain variable fragment of H_2_CasMab-2 has been developed and is currently being evaluated in a phase I clinical trial (NCT06241456) [43]. Although several anti-CD44v6 mAbs clones have been generated, additional clones will be necessary to identify those with cancer-specific reactivity suitable for developing CD44v6-directed CasMabs. We have been generating more clones of anti-CD44v6 mAbs by immunization of CD44v6-overexpressed tumor cells. We have been updated our website antibody bank (http://www.med-tohoku-antibody.com/topics/001_paper_antibody_PDIS.htm, accessed on 20 November 2025). We will screen their reactivity to normal and cancer cells, and the development of anti-CD44v6 CasMabs is expected in the future.

## 5. Conclusions

C_44_Mab-9-mG_2a_, a mouse IgG_2a_ version of a novel anti-CD44v6 mAb, showed a superior reactivity to CD44v6-positive colorectal and gastric cancer cell lines in flow cytometry. C_44_Mab-9-mG_2a_ exerted ADCC, CDC, and antitumor activities in their xenograft models. Our findings indicated that C_44_Mab-9-mG_2a_ could be applied to the mAb-based tumor therapy, and the derivatives such as ADC or CAR-T are expected for future therapy against CD44v6-positive tumors.

## Figures and Tables

**Figure 1 cells-14-01873-f001:**
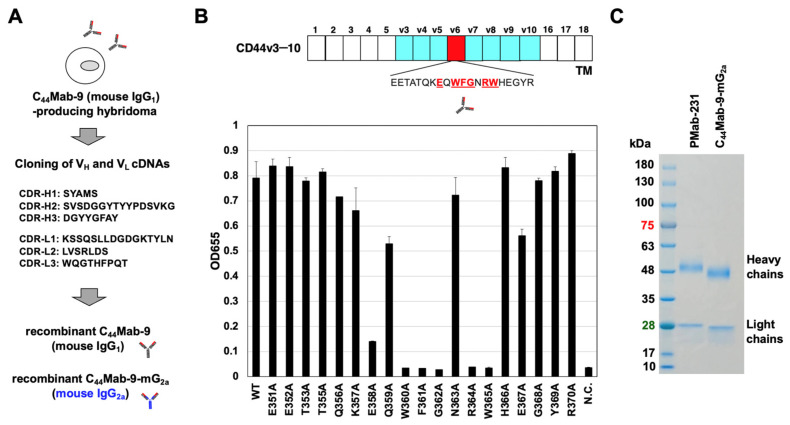
Production of recombinant anti-CD44v6 mAbs, C_44_Mab-9 and C_44_Mab-9-mG_2a,_ from C_44_Mab-9-producing hybridoma. (**A**) After determination of CDRs of C_44_Mab-9 (mouse IgG_1_), recombinant C_44_Mab-9 and C_44_Mab-9-mG_2a_ (mouse IgG_2a_) were produced. The amino acid sequence of V_H_ and V_L_ CDRs were indicated. (**B**) ELISA assay using twenty synthesized CD44v6 peptides [CD44v6 p351–370 wild-type (WT) and the alanine-substituted mutants]. Extracellular structure of CD44v3–10, including standard exons (1–5 and 16–18) and variant exons (v3–v10)-encoded regions are presented. The essential amino acids for C_44_Mab-9 recognition are presented (underlined). N.C.: negative control (PBS). (**C**) C_44_Mab-9-mG_2a_ and PMab-231 (control mIgG_2a_) were subject to SDS-PAGE and the gel was stained with Bio-Safe CBB G-250 Stain.

**Figure 2 cells-14-01873-f002:**
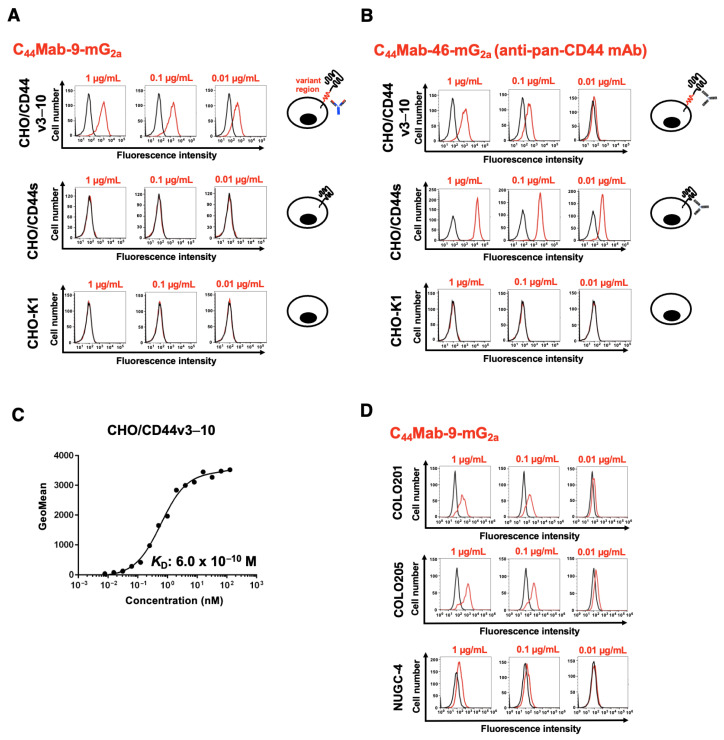
Flow cytometry analysis using anti-CD44 mAbs. (**A**,**B**) CHO/CD44v3–10, CHO/CD44s, and CHO-K1 were treated with 0.01, 0.1, and 1 µg/mL of C_44_Mab-9-mG_2a_ (**A**) and C_44_Mab-46-mG_2a_ (**B**) The cells were treated with Alexa Fluor 488-conjugated anti-mouse IgG. (**C**) CHO/CD44v3–10 were treated with serially diluted C_44_Mab-9-mG_2a_, followed by Alexa Fluor 488-conjugated anti-mouse IgG treatment. The *K*_D_ values were determined. (**D**) COLO201, COLO205, and NUGC-4 were treated with 0.01, 0.1, and 1 µg/mL of C_44_Mab-9-mG_2a_. The cells were treated with Alexa Fluor 488-conjugated anti-mouse IgG.

**Figure 3 cells-14-01873-f003:**
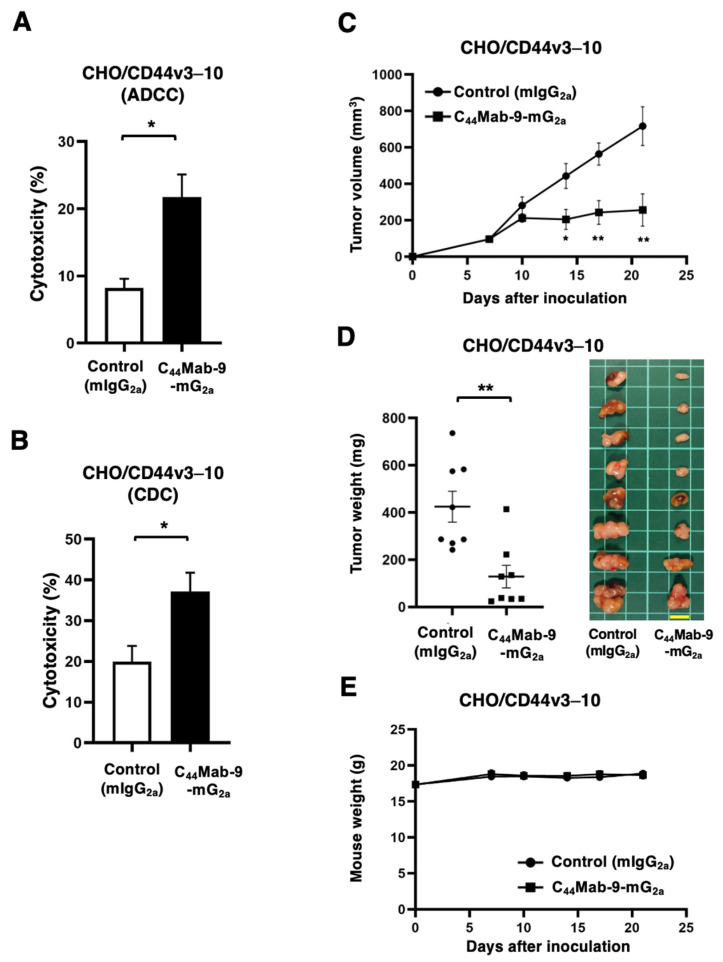
ADCC, CDC, and antitumor effects against CHO/CD44v3–10 by C_44_Mab-9-mG_2a_. (**A**) Calcein AM-labeled target CHO/CD44v3–10 was incubated with the effector splenocytes in the presence of 100 μg/mL of C_44_Mab-9-mG_2a_ or control mIgG_2a_. (**B**) Calcein AM-labeled target CHO/CD44v3–10 was incubated with complements and 100 μg/mL of C_44_Mab-9-mG_2a_ or control mIgG_2a_. Following a 4.5-h incubation, the Calcein release into the medium was measured. Values are shown as mean ± SEM. Asterisks indicate statistical significance (* *p* < 0.05; Two-tailed unpaired *t* test). (**C**–**E**) Antitumor activity of C_44_Mab-9-mG_2a_ against CHO/CD44v3–10 xenograft. (**C**) CHO/CD44v3–10 was subcutaneously injected into BALB/c nude mice (day 0). An amount of 200 μg of C_44_Mab-9-mG_2a_ or control mIgG_2a_ was intraperitoneally injected into each mouse on day 7. Additional antibodies were injected on day 14. The tumor volume (mean ± SEM) is represented. ** *p* < 0.01, * *p* < 0.05 (ANOVA with Sidak’s multiple comparisons test). (**D**) The xenograft weight (mean ± SEM, **left**) and appearance (**right**) are presented. ** *p* < 0.01 (Two-tailed unpaired *t* test). (**E**) Body weight (mean ± SEM) of xenograft-bearing mice treated with the mAbs is presented.

**Figure 4 cells-14-01873-f004:**
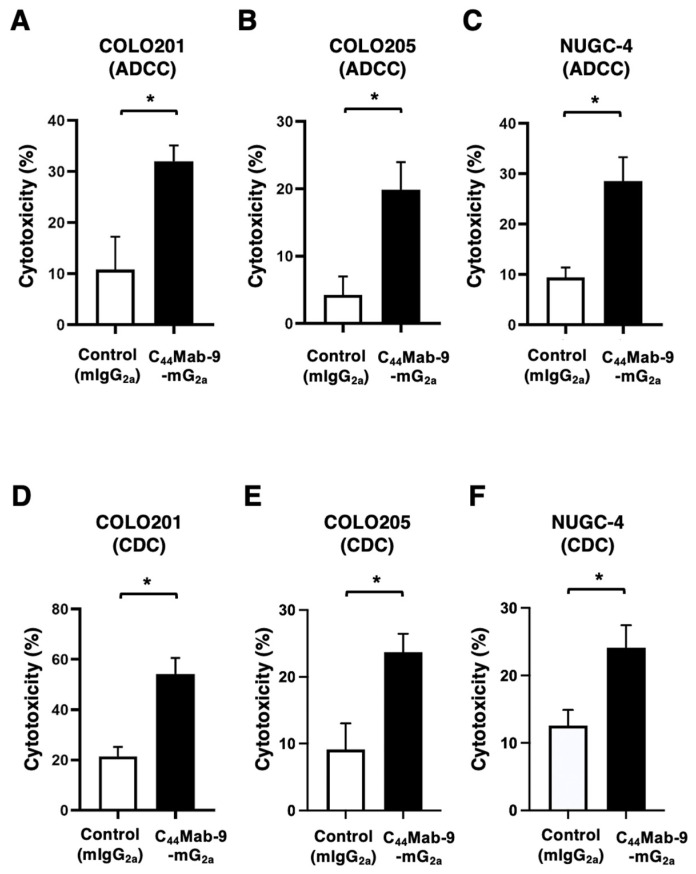
ADCC and CDC by C_44_Mab-9-mG_2a_ against COLO201, COLO205, and NUGC-4. (**A**–**C**) Calcein AM-labeled targets COLO201 (**A**), COLO205 (**B**), and NUGC-4 (**C**) were incubated with the effector splenocytes in the presence of 100 μg/mL of C_44_Mab-9-mG_2a_ or control mIgG_2a_. (**D**–**F**) Calcein AM-labeled targets COLO201 (**D**), COLO205 (**E**), and NUGC-4 (**F**) were incubated with complements and 100 μg/mL of C_44_Mab-9-mG_2a_ or control mIgG_2a_. Following a 4.5-h incubation, the Calcein release into the medium (mean ± SEM) was determined. Asterisks indicate statistical significance (* *p* < 0.05; Two-tailed unpaired *t* test).

**Figure 5 cells-14-01873-f005:**
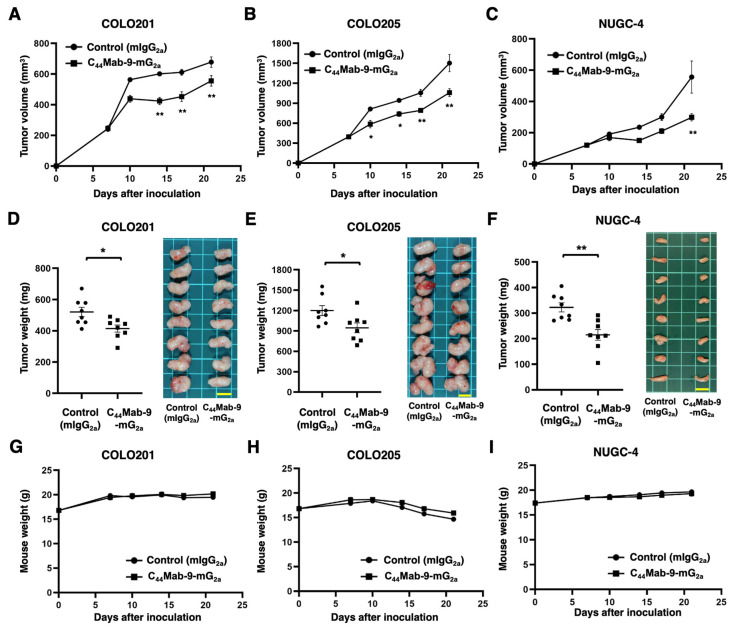
Antitumor activity of C_44_Mab-9-mG_2a_ against COLO201, COLO205, and NUGC-4 xenograft. (**A**–**C**) COLO201 (**A**), COLO205 (**B**), and NUGC-4 (**C**) were subcutaneously injected into BALB/c nude mice (day 0). Amounts of 100 μg (in COLO201 and COLO205) or 200 μg (in NUGC-4) of C_44_Mab-9-mG_2a_ or control mIgG_2a_ were intraperitoneally injected into each mouse on day 7. Additional antibodies were injected on day 14. The tumor volume (mean ± SEM) is represented. ** *p* < 0.01, * *p* < 0.05 (ANOVA with Sidak’s multiple comparisons test). (**D**–**F**). The xenograft weight (mean ± SEM, **left**) and appearance (**right**) are presented. ** *p* < 0.01 (Two-tailed unpaired *t* test). (**G**–**I**) Body weight (mean ± SEM) of xenograft-bearing mice treated with the mAbs is presented.

## Data Availability

The data presented in this study are available in the article.

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
