# Peer review of "Therapeutic Potential of an Anti-CD44v6 Monoclonal Antibody in Xenograft Models of Colorectal and Gastric Cancer"

_cells, 2025, doi:10.3390/cells14231873_

Round 1

Reviewer 1 Report

Comments and Suggestions for Authors

In this manuscript, the authors describe in vivo and in vitro testing of an anti-CD44v6 antibody, which was isolated using mouse immunization and expressed as a recombinant mouse IgG2a antibody. Its epitope was described using Ala-mutated peptides, and found to be overlapping but not completely identical with previously described BIWA4 antibody, which was already tested for human use. They show its binding to diverse cell lines, including CHO transfected with CD44v3–10, and tumor cell lines NUGC-4, COLO201 and COLO205. The antibody could lyse the CHO/CD44v3-10 cells via ADCC or CDC, and inhibited the growth of this cell line in mice. Then the antibody was tested for ADCC and CDC effects on tumor cell lines COLO201, COLO205, and NUGC-4 and was more potent in comparison with an IgG2a with an irrelevant specificity. Its antitumor activity was also in this case shown in vivo, where the growth of COLO201, COLO205, and NUGC-4 xenograft tumors in Balb/c nude mice was diminished when the antibody was applied. Importantly, the weight loss of the affected animals was comparable to controls, indicating low toxicity of the antibody.

The manuscript is well written, the display elements are very systematically prepared, and the discussion section is interesting to read. The potential of already tested anti-CD44 therapeutic agents is critically discussed. This is the main point that I would recommend to address in the Discussion: what is the strategy of the authors to develop their anti-CD44v6 antibody, to avoid potential safety issues and lack of effectivity (e.g. bispecific antibody development, toxin or radioconjugate…) – at the moment, only humanization is mentioned as a next planned step. The ADCC and CDC experiments in vitro do not show the effect of the antibody on an antigen-negative cell lines, this data should be added.

Please find below a list of remarks which I hope will be helpful.

Line 23: „applies to multiple experimental techniques“ – very general sentence and more specific information would be helpful (are this mechanisms of action, diagnostic techniques…)

Line 27: overexpressing

Line 139:” Alexa Fluor 488-conjugated anti-mouse IgG“ – all commercially acquired antibodies should be specified with RRID-identifiers.

Line 293: complement, not complements

Line 319: “Therefore, targeting CD44v6 has been considered as an attractive target for mAb-based therapy“ – Therefore, CD44v6…

Line 329: …was developed within the context of CD44v6-directed CAR-T

Line 332: overlap with the epitope sequence

Line 381: “using some strategy“ – please be more specific

Reviewer 2 Report

Comments and Suggestions for Authors

Congratulations, this is solid work, just needs a bit of "tempering" the interpretations. The experimental design, and all the different methods used are logical and well structured. It progrsses step by step in a logical path, starting from epitope mapping to 

The experimental design is logical and well-structured, progressing from the incentive (CD44v6 as a target) to v6 epitope mapping, followed thorough analysis of the binding properties, and equally thorough analysis of in vitro cytotoxicity- - and culminates eventually in animal experiments measuring in vivo toxicity and reduction of tumor growth.  It should be mentioned that proper, useful controls were used throughout and that not one, but multiple colorectal cancer cell lines were used, improving the scientific quality of the data and the manuscript as a whole. Both antibody-dependent cytotoxicity and complement-dependent cytotoxicity were analyzed (ADCC & CDC). Beyond that, however, no additional mechanisms going deeper have been analyzed. What exactly is the functional basis for this observed cytotoxicity? Which mechanisms/cellular pathways are affected? 

The antibody under investigation is a mouse IgG2a antibody, which limits its translational value, but the authors do mention they are preparing a human/humanized version for subsequent testing. HGF signaling is discussed, but not shown / tested although its fundamental for the functions of CD44 and especially CD44v6: "The v6-encoded region is critical for activation of c-MET via a ternary complex formation with the ligand, hepatocyte growth factor (HGF)". 

as far as know, the statistical methods used here are simple and appropriate (ANOVA, with t-tests), the statistical power appears appropriate too (although theres no mention of this in the manuscript). Error bars (SEM) appear at the right places throughout the figures and in the text. heres only 3 replicates for the in vitro studies, compared to 8 animals in the groups compared  in xenograft studies. The cytotoxic effects observed are not exactly dramatic but they seem to be statistically significant... so all is well. Nevertheless, based on these rather modest effects in animal/xenograft studies - the conclusions appear a bit bold. there is only between 20 - 30% reduction with 3 of the CRC cell lines, which is not exactly "potent" as stated by the authors... statistically significant, yes - but not necessarily "potent" . 

  •  

Therefore also some of the discussion is somewhat speculative, such as the topic related to ferroptosis induction (which is not shown in the paper), but should be discussed in depth without any supporting evidence, other than maybe from the literature ... were not exactly dealing with "cancer-specific monoclonal antibodies" as stated. Everyone wants "CasMabs" but its not so easy, it seems. There are also a good deal of self-citation (e.g. the references 20-30, 32, and 46). ) -  but at least, they also seem to be quite appropriate, this should be balanced

Technically, a few issues could have complemented the paper nicely, such as any comparison of the self-made mAvs with commercial ones with existing anti-CD44v6 antibodies (like VFF18, which is commercially available, and BIWA-4 Bivatuzumab mentioned in the literature ) in parallel experiments. Tsting iif HGF/c-Met are targeted wold also be nice. 

I specifically like the figures: it is the number of figures that is suitable for a "classic" publication (no more than 6 figures in total), and even more importantly, none of the 5 figures is overloaded with subfigures  - they are straightforward, also easy to compregend and the figure legends are matching the content of the figures and is neither too detailed nor lacks critical information. 

Reviewer 3 Report

Comments and Suggestions for Authors

The paper is generally rigorous in terms of experimental design, data presentation, and scientific validity, but there are still some issues and room for improvement.

  1. In introduction, how about the potential toxicity of CaMab-9-mG2a in humans;
  2. No CDR sequence was provided. Please analyze antibody binding mechanisms deeply.
  3. Does C44Mab-9 compete with VFF18 to bind to the same region?
  4. Are there any differences in expression levels and glycosylation between CHO cells overexpressing CD44v3-10 and human tumor cells expressing endogenous CD44v6?
  5. The sample size of animal experiments is relatively small; please specify whether random grouping or blind evaluation was conducted.
  6. No immune response indicators such as cytokine release or T cell activation were detected. Please supplement related experiments or provide reasons.
  7. Please compare the study subject horizontally with VFF18 or other clinical stage antibodies.
  8. Please verify whether ferroptosis is induced, although this possibility was mentioned in the discussion.
  9. Does C44Mab-9 affect the c-MET signaling pathway mediated by CD44v6?
  10. The chart is clear, but lacks specific identification for statistical testing (such as p-values labeled in the chart).
  11. Some reference formats are not consistent, it is recommended to use the format required by the journal uniformly throughout the entire text.

Round 2

Reviewer 3 Report

Comments and Suggestions for Authors
  1. the title should be changed to a complex phrases rather than a sentence ("exhibits");
  2. The SDS-PAGE conditions for Fig.1C are not provided;
  3. The caption of Fig.3d does not cover the tumor photo (right), and the latter can be larger (also for Fig.5d,e,f).
  4. The significance, highlights, features and contribution should be further strengthened in the part of Conclusion.

Author Response

the title should be changed to a complex phrases rather than a sentence ("exhibits");

--> We changed.

The SDS-PAGE conditions for Fig.1C are not provided;

-->We added in section 2.2.

The caption of Fig.3d does not cover the tumor photo (right), and the latter can be larger (also for Fig.5d,e,f).

-->We added the explanation in the legends.

The significance, highlights, features and contribution should be further strengthened in the part of Conclusion.

-->We revised.